



**Along-stream transport and transformation of dissolved organic**
**matter in a large tropical river**
Thibault Lambert[1,*], Cristian R. Teodoru[2], Frank C. Nyoni[3], Steven Bouillon[2], François
Darchambeau[1], Philippe Massicotte[4] and Alberto V. Borges[1].
[1] University of Liège, Chemical Oceanography Unit, Liège, Belgium
[2] KU Leuven, Department of Earth and Environmental Sciences, Leuven, Belgium
[3] University of Zambia, Integrated Water Resources Management Center, Lusaka,
Zambia
[4] Aarhus University, Department of Bioscience, Denmark
* Corresponding author



**Abstract -** Large rivers transport considerable amounts of terrestrial dissolved organic matter (DOM) to the ocean. Yet, downstream gradients and temporal variability in DOM fluxes and characteristics are poorly studied at the scale of large river basins, especially in tropical areas. Here, we report longitudinal patterns in DOM content and composition based on absorbance and fluorescence measurements along the Zambezi River and its main tributary, the Kafue River, during two hydrological seasons. During high flow periods, a greater proportion of aromatic and humic DOM was mobilized along rivers due to the hydrological connectivity with wetlands and high flow velocities, while low flow periods were characterized by lower DOM content of less aromaticity resulting from loss of connectivity with wetlands, more efficient degradation of terrestrial DOM and enhanced autochthonous productivity. Changes in water residence time due to contrasting water discharge were found to modulate the fate of DOM along the river continuum. Thus, terrestrial DOM dynamics shifted from transport-dominated during the wet seasons towards degradation during the dry season, with substantial consequences on longitudinal DOM content and composition. The longitudinal evolution of DOM was also strongly impacted by a hydrological buffering effect in large reservoirs in which the seasonal variability of DOM fluxes and composition was strongly reduced.



## 1. Introduction

The composition, transport and transformation of dissolved organic matter (DOM) in large rivers are key aspects for determining regional and global carbon (C) budgets (Schlesinger and Melack, 1981), the fate of terrigenous DOM flowing to the oceans (del Giorgio and Pace, 2008; Massicotte and Frenette, 2011), the influence of fluvial inputs on DOM biogeochemistry in coastal and oceanic environments (Holmes et al., 2008), and the functioning of inland waters as active pipes with regards to the global C cycle (Cole et al., 2007; Borges et al., 2015a). Riverine DOM is mainly derived from terrestrial soils (e.g. Weyhenmeyer et al., 2012), but can also be fueled by sources within the aquatic system (Lapierre and Frenette, 2009; Massicotte and Frenette, 2011). Longitudinal patterns of riverine DOM, both in terms of concentration and composition, are controlled by numerous environmental drivers including connectivity with surrounding wetlands (Battin, 1998; Mladenov et al., 2007), lateral inputs from tributaries (Massicotte and Frenette, 2011) and shifts in dominant land cover (Ward et al., 2015). Once in the aquatic ecosystem, terrestrial DOM is exposed to in-stream processing such as photodegradation (Cory et al., 2007; Spencer et al., 2009), microbial respiration (Amon and Benner, 1996; Fasching et al., 2014), and flocculation (von Wachenfeldt and Tranvik, 2008), that usually operate simultaneously and lead to the removal and the transformation of DOM during its transport (Massicotte and Frenette, 2011; Cawley et al., 2012). The overall reactivity of DOM in freshwater is largely controlled by its composition (Kothawala et al., 2014; Kellerman et al., 2015). For example, the selective loss of the colored fraction of terrestrial DOM is a common pattern observed in a wide variety of ecosystems (Moran et al., 2000; Cory et al., 2007; Spencer et al., 2009; Weyhenmeyer et al., 2012). However, the extent of DOM decay depends on the water residence time (WRT) of the aquatic ecosystem (Cory et al.,



2007; Hanson et al., 2011; Köhler et al., 2013). In large rivers, WRT varies spatially,
increasing in reservoirs and lakes compared to river channels, and seasonally, being
higher during low flow compared to high flow. Considering that changes in water level also
control the hydrological connectivity with wetlands, it is likely that the downstream gradient
in DOM composition drastically differs in relation to spatial and temporal changes in
hydrodynamic conditions.
Longitudinal patterns of DOM in large rivers are often assessed in one specific
environment, such as wetlands/floodplains (Mladenov et al., 2007; Yamashita et al., 2010;
Cawley et al., 2012; Zurbrügg et al., 2013) or lakes (Parks and Baker, 1997; Massicotte
and Frenette 2013; Stackpoole et al., 2014), or limited to a subsection of large rivers (del
Giorgio and Pace; 2008; Massicotte and Frenette, 2011; Ward et al., 2015), and mostly
carried out during one specific hydrological period. Our understanding of rivers as a
continuum in which DOM is simultaneously transported from terrestrial soils to oceans,
produced and degraded is thus fundamentally limited by a lack of basin-scale studies
taking into account seasonal variations. This is especially true for tropical waters that have
the highest riverine dissolved organic carbon (DOC) flux to the oceans (Meybeck, 1993)
but for which DOM cycling has received less attention than rivers in other biomes with the
exception of the Amazon River (Mayorga et al., 2005; Johnson et al., 2011; Ward et al.,

2013; 2015).

The study of DOM biogeochemistry at large spatial and temporal scales requires
analytical tools that are simple to implement but have a large sample throughput while
providing pertinent information about the DOM chemical composition. Spectroscopic
methods, primarily based on ultraviolet-visible and fluorescence measurements, fulfill
these criteria (Jaffé et al., 2008). Optical properties of colored DOM (CDOM) and




fluorescent DOM (FDOM) can be used to calculate several indices related to DOM
composition. These include the specific ultra violet absorbance at 254 nm ($SUVA_{254}$),
positively related to the degree of DOM aromaticity (Weishaar et al., 2003), the spectral
slope ratio ($S_R$), inversely related to the average DOM molecular weight (Helms et al.,
2008) and the fluorescence index (FI), related to the contribution of terrestrial versus
microbial inputs (McKnight et al., 2001). FDOM measurements acquired as three-
dimensional excitation-emission matrices (EEMs) and coupled with the parallel factor
analysis (PARAFAC) provide additional benefits for the characterization of DOM
(Stedmon et al., 2003; Stedmon and Markager, 2005; Yamashita et al., 2008). In addition,
the carbon stable isotope composition of DOC ($\delta^{13}C_{DOC}$) can provide information about
the terrestrial or aquatic origin of DOM (Mladenov et al., 2007; Lambert et al., 2015).

The Zambezi River basin, the fourth largest river in Africa, was extensively sampled

from its source to its mouth during three field campaigns carried out over wet and dry
seasons (Teodoru et al., 2015; Fig. 1 and 2). Longitudinal patterns of DOM were assessed
through measurements of DOC concentrations and characterization of DOM ($\delta^{13}C_{DOC}$
coupled with CDOM and FDOM) along the Zambezi River (>3000 km) and its main
tributary, the Kafue River (>1500 km). The aim of this study was to determine the main
drivers on downstream patterns of DOM at the scale of a large tropical river, with a specific
attention for the role of WRT in modulating the fate of DOM.
**2. Materials and methods**
**2.1. Study site.** The Zambezi River has a drainage area of $1.4 \times 10^6$ km², originates in
northwest Zambia and flows southeast over 3000 km before it discharges into the Indian
Ocean in Mozambique (Fig. 1). The climate of the Zambezi Basin is classified as humid



subtropical and is characterized by two main seasons, the rainy season from
October/November to April/May and the dry season from May/June to
September/October. Annual precipitation strongly varies with latitude, from > 2000 mm in
the northern part and around Lake Malawi to less than 500 mm in the southern part of the
basin. The mean annual rainfall over the entire catchment is ~940 mm (Chenje, 2000). Up
to 95% of the annual rainfall occurs during the rainy period while the dry period presents
irregular and sporadic rainfall events. Consequently, water discharge in Zambezi River
has a bimodal distribution with a single maximum peak discharge occurring typically in
April/May and a minimum in November (Fig. 2).

Woodlands and shrublands are the dominant (55%) land cover and stretch over the

whole catchment, forests (20%) and grasslands (9%) areas are mainly confined to the
northeast part of the basin and croplands represents 13% of the total area (Mayaux et al.,
2004). Wetlands, including swamps, marshes, seasonally inundated floodplains and
mangroves cover 5% of the total basin area (Lehner and Döll, 2004).

Based on distinct geomorphological characteristics, the Zambezi Basin can be divided

into three major segments: (1) the upper Zambezi from the headwaters to Victoria Falls;
(2) the middle Zambezi, from Victoria Falls to the edge of the Mozambique coastal plain
(below Cahora Bassa Gorge); and (3) the lower Zambezi, the stretch crossing the coastal
plain down to the Indian Ocean (Wellington, 1955). The upper Zambezi covers about 40%
of the total area of the Zambezi basin but comprises the highest fraction of wetlands and
floodplains (about 60% of the total wetlands/floodplains areas of the Zambezi Basin),
including the Barotse Floodplain and the Chobe Swamps (Fig. 1). The middle stretch of
the Zambezi River is buffered by two major man-made impoundments, namely the Kariba
Reservoir (volume: 157 km³; area: 5364 km²) and the Cahora Bassa Reservoir (volume:



63 km³; area: 2739 km²). The Kafue River (drainage area: $1.56 \times 10^5$ km²) joins the
Zambezi River $\sim$ 70 km downstream of the Kariba Dam. Similarly to the upper Zambezi,
the Kafue River comprises a high density of wetlands/floodplains (about 26% of the total
wetlands/floodplains areas of the Zambezi basin), including the Lukanga Swamps and the
Kafue Flats (Fig. 1). It also comprises two smaller reservoirs, the Itezhi Tezhi Reservoir
(volume: $\sim$ 6 km³; area: 365 km²) and the Kafue Gorge Reservoir (volume: $\sim$1 km³; area:
13 km²). In its lower part, the Zambezi River and its tributary the Shire River both drain
narrow but $\sim$ 200 km long wetlands areas before their confluence zone. At the end of its
course, the river forms a large, 100 km long floodplain-delta system of swamps and
meandering channels.
**2.2. Sampling and analytical methods.** Sampling was conducted during two
consecutive years and over two climatic seasons: wet season (1 February to 5 May, n=40)
2012, wet season (6 January to 21 March, n=41) 2013, and dry season (15 October to 28
November, n=24) 2013 (Fig. 2). Sites in the Zambezi and the Kafue rivers were located
100 – 150 km apart from the spring to the outlet (Fig. 1) except during the 2013 dry season
when sampling in the Zambezi River ended before its entrance in the Cahora Bassa
Reservoir due to logistical constraints.
Water sampling was mainly performed from boats or dugout canoes in the middle
of the river. In few case (n=10), in the absence of boats/canoes, sampling was carried out
either from bridges or directly from the shore and as far as possible away from the
shoreline, but without discernable effects on the longitudinal patterns on DOM or other
biogeochemical variables (Teodoru et al., 2015). Approximately 2 L of water were
collected 0.5 m below the surface, kept away from direct sunshine and filtered and
conditioned within 2 h of sampling. Filtrations were performed successively on pre-





combusted GF/F glass fiber filters (0.7 µm porosity), then on 0.2 µm polyethersulfone
syringe filters. Samples for the measurement of DOC concentration and $\delta^{13}C_{DOC}$
signatures were stored in 40 mL glass vials with polytetrafluoroethylene (PTFE) coated
septa with 50 µL $H_3PO_4$ (85%). Samples for CDOM/FDOM analyses were stored in 20 mL
amber glass vials with PTFE-coated septa but without $H_3PO_4$ addition. Samples for major
elements (including Fe) were stored in 20 mL scintillation vials and acidified with 50 µl of
HNO3 65 % prior to analysis.
**2.3. DOC analysis.** DOC and $\delta^{13}C_{DOC}$ were analyzed with an Aurora1030 total organic
carbon analyzer (OI Analytical) coupled to a Delta V Advantage isotope ratio mass
spectrometer. Typical reproducibility observed in duplicate samples was in most cases <
± 5 % for DOC, and ± 0.2 ‰ for $\delta^{13}C_{DOC}$. Quantification and calibration was performed
with an aqueous solution of IAEA-C6 and in-house sucrose standards.
**2.4. CDOM analysis and calculations.** Absorbance was recorded on a Perkin-Elmer
UV/Vis 650S spectrophotometer using a 1 cm quartz cuvette. Absorbance spectra were
measured between 190 and 900 nm at 1 nm increment and instrument noise was
assessed measuring ultrapure (Type 1) Milli-Q (Millipore) water as blank. After subtracting
the blank spectrum, the correction for scattering and index of refraction was performed by
fitting the absorption spectra to the data over the 200-700 nm range according to the
following equation:

$$A_\lambda = A_0 e^{-S(\lambda - \lambda_0)} + K \qquad\qquad (1)$$

where $A_\lambda$ and $A_0$ are the absorbance measured at defined wavelength $\lambda$ and at reference
wavelength $\lambda_0 = 375$ nm, respectively, S the spectral slope ($nm^{-1}$) that describes the
approximate exponential decline in absorption with increasing wavelength and K a



background offset. The fit was not used for any purpose other than to provide an offset
value K that was then subtracted from the whole spectrum (Lambert et al., 2015).

The $SUVA_{254}$ was calculated as the UV absorbance at $\lambda = 254$ nm ($A_{254}$) normalized

to the corresponding DOC concentration (Weishaar et al., 2003). The natural UV
absorbance of Fe at $\lambda = 254$ nm was estimated based on measured Fe concentrations
and was then subtracted from the UV absorbance measured. The corrected value of $A_{254}$
was then used to calculate $SUVA_{254}$. The $SUVA_{254}$ was used as an indicator of the
aromaticity of DOC with high values ($>3.5$ I mgC$^{-1}$ m$^{-1}$) indicating the presence of more
complex aromatic moieties and low values ($<3$ I mgC$^{-1}$ m$^{-1}$) indicative the presence of
mainly hydrophobic compounds (Weishaar et al., 2003).

Napierian absorption coefficients were calculated according to:

$$a_\lambda = 2.303 \times A_\lambda/L \qquad\qquad (3)$$

where $a_\lambda$ is the absorption coefficient (m$^{-1}$) at wavelength $\lambda$, $A_\lambda$ the absorbance corrected
at wavelength $\lambda$ and L the path length of the optical cell in m (0.01 m). CDOM was reported
as the absorption coefficient at 350 nm ($a_{350}$). Spectral slopes for the intervals 275-295
nm and 350-400 nm were determined from the linear regression of the log-transformed $a$
spectra versus wavelength. The slope ratio $S_R$ was calculated as the ratio of $S_{275-295}$ to
$S_{350-400}$ according to Helms et al. (2008). $S_R$ is related to the molecular weight distribution
of DOM with values less than 1 indicative of enrichment in high molecular weight
compounds and high values above 1 indicative of a high degree of low molecular weight
compounds (Helms et al., 2008).
**2.5. FDOM analysis and PARAFAC modeling.** Fluorescence intensity was recorded on
a Perkin-Elmer LS55 fluorescence spectrometer using a 1 cm quartz cuvette across
excitation wavelengths of 220-450 nm (5 nm increments) and emission wavelengths of



230-600 nm (0.5 nm increments) in order to build excitation–emission matrices (EEMs). If necessary, samples were diluted until $A_{254}$ < 0.2 m$^{-1}$ to avoid problematic inner filter effects (Ohno, 2002). Before each measurement session (i.e. each day), a Milli-Q water sample was also analysed. EEMs preprocessing such as removing first and second Raman scattering, standardization to Raman units, absorbance corrections and inner filter effects were performed prior the PARAFAC modelling. The scans were standardized to Raman's units (normalized to the integral of the Raman signal between 390 nm and 410 nm in emission at a fixed excitation of 350 nm) with a Milli-Q water sample run the same day as the samples (Zepp et al., 2004). PARAFAC model was using MATLAB (MathWorks, Natick, MA, USA) and DOM Fluorescence Toolbox 1.7. PARAFAC model was validated by split-half analysis and random initialization (Stedmon and Bro, 2008). Additional samples analysed in the same manner and collected from (1) tributaries of the Zambezi and the Kafue rivers as well as during a two-years monitoring period of the Zambezi and the Kafue rivers (n = 42; data not published), and (2) the Congo Basin (n = 164; data not published) were added to the dataset. This was done to increase the variability of DOM fluorescence signatures and therefore help detect components that could have been present in insufficient quantity to be detected in our environment (Stedmon and Markager, 2005). The maximum fluorescence $F_{Max}$ values of each component for a particular sample provided by the model were summed to calculate the total fluorescence signal $F_{Tot}$ of the sample in Raman's unit (R.U.). The relative abundance of any particular PARAFAC component X was then calculated as $\%C_X = F_{Max}(X)/F_{Tot}$. The FI index was calculated as the ratio of the emission intensities at 470 nm and 520 nm at an excitation wavelength of 370 nm (McKnight et al., 2001). A higher FI value (e.g., 1.8) indicates a microbial DOM




source while a lower value (e.g., 1.2) indicates a terrestrial source; intermediate values
indicate a mixed DOM source.
**2.6. Statistical Analysis**
PCA was performed on scaled variables using the prcomp function in R software. DOC
concentrations, stable carbon isotopic composition, optical indices (SUVA$_{254}$, S$_R$, FI), a$_{350}$,
$F_{Max}$ and the relative abundance of PARAFAC components were used as the variables for
the PCA. Given the different units of the variables used in the PCA, data were scaled to
zero-mean and unit-variance as recommended (Borcard et al., 2011). The PCA was then
performed on the correlation matrix of the scaled variables.
**3. Results**
**3.1. Longitudinal patterns in DOC concentration, composition and DOM optical**
**properties**

Data were acquired during two wet seasons and one dry season, the two wet

seasons data are discussed together hereafter. DOC concentrations in the Zambezi River
ranged from 1.9 ± 0.1 to 4.9 ± 1.0 mg L$^{-1}$ during the wet periods and from 1.2 to 2.9 mg L$^{-1}$
$^1$ and the dry period (Fig. 3A). Along the upper Zambezi DOC increased downstream
during the wet seasons, while DOC gradually decreased downstream during the dry
season. In the Kariba Reservoir, DOC variability between wet and dry seasons was
relatively low, and concentrations ranged from 2.4 ± 0.3 to 2.9 ± 1.4 mg L$^{-1}$. DOC exhibited
relatively small variability downstream of the Kariba Reservoir and along the lower
Zambezi, with the exception of a slight increase during the wet seasons downstream of
the confluence with the Shire River (outlet of Lake Malawi).



In the Kafue River, DOC was generally higher during the wet seasons (from $3.1 \pm$
$0.1$ to $5.4 \pm 0.7$ mg $L^{-1}$) compared to the dry season (from 1.3 to 3.6 mg $L^{-1}$)(Fig. 3B).
Despite this seasonal difference, DOC increased gradually downstream during both wet
and dry seasons. DOC concentrations in the Itezhi Tezhi Reservoir showed a decrease
(~25%) during the wet seasons but an increase (~20%) during the dry season compared
to the upstream station.

The $a_{350}$ values (Fig. 3C and 3D) were higher during the wet seasons (1.7 to 16.6

$m^{-1}$ in the Zambezi and 3.9 to 11.5 $m^{-1}$ in the Kafue) than during the dry season (1.3 to
10.7 $m^{-1}$ in the Zambezi and 1.2 to 4.7 $m^{-1}$ in the Kafue). They followed similar spatial and
seasonal patterns as DOC concentrations, with some differences. First, decreases in $a_{350}$
values were more pronounced than for DOC, especially in the upper Zambezi during the
dry season and in the Kariba and Itezhi Tezhi reservoirs during the wet season. For
example, while DOC decreased by a factor ~2 as the Zambezi enters the Kariba Reservoir
during the wet periods, $a_{350}$ decreased by a factor ~4. Secondly, while DOC
concentrations were higher at the outlet of reservoirs compared to upstream stations
during the dry season, $a_{350}$ values were lower.

$\delta^{13}C_{DOC}$ showed a gradual increase along the Zambezi River during all periods

from -28.1 and -26.5 ‰ at the source to -21.4 to -20.1 ‰ near its delta, the latter being
especially marked between the two first sampling sites in the upper Zambezi (Fig. 3E),
while no significant pattern was observed along the Kafue River (values between -25.9
and -20.5 ‰, Fig. 3F).

DOM at the source of the Zambezi exhibited the highest SUVA$_{254}$ (> 4 L mgC$^{-1}$ m$^{-1}$)

$^1$) and lowest $S_R$ (< 0.8) values during both wet and dry seasons (Fig. 3G and 3I). During
the wet seasons, the upper Zambezi was characterized by stable SUVA$_{254}$ (3.5 – 4.0 L



mgC$^{-1}$ m$^{-1}$) and low S$_R$ (0.85 – 0.91) values. In the middle Zambezi, SUVA$_{254}$ and S$_R$
values were lowest (2.2 ± 0.2 – 2.9 ± 0.1 L mgC$^{-1}$ m$^{-1}$) and highest (1.22 ± 0.09 – 1.41 ±
0.01) in the Kariba and the Cahora Bassa reservoirs compared to samples collected in-
between (2.6 ± 0.1 – 3.1 ± 0.02 L mgC$^{-1}$ m$^{-1}$ for SUVA$_{254}$ and 0.97 ± 0.1 – 1.10 ± 0.08 for
S$_R$). Overall, SUVA$_{254}$ increased from 2.1±0.5 to 2.9±0.9 L mgC$^{-1}$ m$^{-1}$ whereas S$_R$
decreased from 1.08±0.09 to 0.97±0.04 in the lower Zambezi, with a maximum (3.3±0.9
L mgC$^{-1}$ m$^{-1}$) and a minimum (0.88±0.006) values recorded below the confluence with the
Shire River, respectively. During the wet periods, FI values ranged between 1.24 and 1.41
in the mainstream, and between 1.43 and 1.58 in reservoirs (Fig. 3K). FI values during
the dry season were globally higher than during the wet periods with values ranging from
1.29 to 1.72, expect at the source of the Zambezi, where an FI value of 1.19 was observed.
In the Kafue River, variations in DOM composition were marked between the wet
and dry seasons, but minimal along the longitudinal transect (Fig. 3H, 3J and 3L). SUVA$_{254}$
and S$_R$ ranged from 3.5 to 4.0 L mgC$^{-1}$ m$^{-1}$ and from 0.79 to 1.05, respectively, during the
wet seasons, except in the Itezhi Tezhi Reservoir where SUVA$_{254}$ decreased to 2.4 L mgC$^{-}$
$^1$ m$^{-1}$ and S$_R$ increased up to 1.16. Values were quite stable during dry periods, and ranged
between 2.2 and 2.8 L mgC$^{-1}$ m$^{-1}$ for SUVA$_{254}$ and from 1.11 to 1.22 for S$_R$. FI values
ranged between 1.27 and 1.42 during the wet seasons, and between 1.41 and 1.74 during
the dry season.
**3.2. Longitudinal patterns in FDOM**
PARAFAC modelling identified three terrestrial humic-like components (C1, C2 and
C4), one microbial humic-like component (C3) and one protein tryptophan-like (C5)
component (Table 1 and Supplementary Fig. 1). In the Zambezi River, the fluorescence
intensities ($F_{Max}$) of PARAFAC components during the wet seasons presented patterns



similar to DOC concentrations with some exceptions (Fig. 4). The increase of $F_{Max}$ for the
C4 component (calculated as the percentage of increase between lowest and highest
values recorded in corresponding river sections, data not shown) was higher than for the
other components in river sections draining wetlands/floodplains in the upper and lower
Zambezi. All terrestrial and microbial humic-like components showed a systematic and
marked decrease in their $F_{Max}$ values in reservoirs, while $F_{Max}$ of C5 decreased in a smaller
proportion in the Kariba Reservoir and increased in the Cahora Bassa Reservoir. During
the dry season, $F_{Max}$ of terrestrial humic-like components decreased downstream as DOC
concentrations, while $F_{Max}$ remained stable for C3 or increased for C5. In the Kafue River,
$F_{Max}$ of all components followed similar spatial and temporal patterns as those of DOC
concentrations. The main difference observed was that while $F_{Max}$ values of humic-like
compounds were lower during the dry season compared to the wet seasons, $F_{Max}$ of C5
exhibited similar values accross the hydrological cycle.
As a direct consequence of the spatial and temporal differences in $F_{Max}$ of
PARAFAC components, the relative contribution of each component to the total
fluorescence signal $F_{TOT}$ showed distinct patterns (Fig. 5). Thus, the downstream
decrease of %C1 and %C2 observed in the upper Zambezi during the wet seasons can
be related to the parallel increase of %C4, the latter being due to the more pronounced
increase in $F_{Max}$ of C4 relative to the other components. The same patterns for %C1 and
%C2 observed during the dry season, however, reflect the fact that $F_{Max}$ values of C3 and
C5 were stable or increased during the dry season, respectively, while $F_{Max}$ of C1 and C2
decreased. %C5 was higher during the dry season compared to the wet seasons, and
reached highest values in reservoirs during the wet periods due to its specific spatial and
temporal variations in $F_{Max}$ values. No longitudinal changes in the relative abundance of





PARAFAC components were observed along the Kafue River. Similar to what was
observed along the Zambezi River, the dry season was marked by a decrease in %C4
and an increase in %C5, while %C1, %C2 and %C3 were equivalent to values recorded
during the wet seasons.
**3.3. Principal component analysis (PCA)**
The first two components of the PCA explained 71.7% of the variance and
regrouped the variables in three main clusters (Fig. 6). The first includes %C1, %C2 and
samples collected at or near the source of the Zambezi. The second group was defined
by %C4 and several variables including DOC, $F_{Max}$, $SUVA_{254}$ and $a_{350}$. Samples from the
upper Zambezi and from the Kafue rivers (excluding reservoirs) were mainly located in
this cluster. Finally, %C3 and %C5 were clustered with $S_R$ and FI. Samples from reservoirs
(including Kariba, Cahora Bassa and Itezhi Tezhi) were almost all in this cluster. Samples
from the middle and lower Zambezi collected during the wet seasons and those collected
during the dry season were located between the distinct clusters defined by PARAFAC
components and other variables.
**4. Discussion**
**4.1. Identification of PARAFAC components**. Humic-like components C1 and C2 are
among the most common fluorophores found in freshwaters and are associated with high
molecular weight and aromatic compounds of terrestrial origin (Stedmon and Markager,
2005; Yamashita et al., 2008; Walker et al., 2013). Component C4 has been reported to
be of terrestrial origin (Stedmon and Markager, 2005; Kothawala et al., 2015) or to be a
photoproduct of terrestrially derived DOM (Massicotte and Frenette, 2011). The
association of %C4 with DOC concentrations and terrestrial optical indices including $a_{350}$





and SUVA$_{254}$ advocates for a terrestrial origin of this component (Fig. 6). Inversely, %C3
and %C5 were negatively correlated with a$_{350}$ and SUVA$_{254}$. C3 and C5 components are
respectively classified as microbial humic-like and tryptophan-like components related to
the production of DOM within aquatic ecosystems (Kothawala et al., 2014; Kellerman et
al., 2015). Both fluorophores can originate from autochthonous primary production
(Yamashita et al., 2008; 2010; Lapierre and Frenette, 2009) or from degradation of
terrestrial DOM in the water column as previously reported in a wide variety of
environments as marine (Jørgensen et al., 2011) and lake waters (Kellerman et al., 2015)
for C3, and large Arctic rivers (Walker et al., 2013) or small temperate catchment (Stedmon
and Markager, 2005) for C5. The opposite relationship of %C1 and %C2 versus %C3 (Fig.
6) suggests that C3 would be the result of the transformation of terrestrial components C1
and C2 through biological activity in the water column as suggested by Jørgensen et al.
(2011). The distribution of samples along PC1 is thus likely controlled by the transition
from terrestrial DOM with a high degree of aromaticity and humic content (negative
loadings) to less aromatic DOM produced within the aquatic ecosystem by the degradation
of terrestrial DOM during transport and/or by autochthonous sources (positive loadings).
**4.2. Seasonal and spatial variability in downstream gradients in DOM concentration**
**and composition.** Altogether data showed clear changes in the downstream gradients of
DOM concentration and composition, both seasonally and spatially. These changes were
essentially controlled by three main factors: WRT and connectivity with
wetlands/floodplains, both highly dependent on seasonal variations of water level (and
discharge), and water retention by lakes/reservoirs that is more independent from
seasonal variations of water level. Dominant land cover was also found to affect DOM
gradients, but to a lesser degree.



**4.2.1 Land cover and hydrological connectivity with wetlands/floodplains.** The DOM at the source of the Zambezi was clearly distinct from the rest of the basin, independently of the hydrological period (Fig. 6), with a strong aromatic character (highest $SUVA_{254}$), a high degree of molecules with elevated molecular weight (lowest $S_R$) and low $\delta^{13}C_{DOC}$. The dominant land cover quickly shifts from forest in the northern part of the basin where the Zambezi takes its source to grassland and woodland/shrubland that dominate in the rest of the basin (Supplementary Fig. 2). This shift in land cover was reflected in the DOM gradient from the source station of the Zambezi to the next sampling site, and marked by an increase in $S_R$, $\delta^{13}C_{DOC}$ and a decrease in $SUVA_{254}$. This pattern is consistent with the role of forest in releasing more aromatic DOM of high molecular weight than other vegetation types in tropical freshwaters (Lambert et al., 2015).

Downstream, the variability in the optical properties of DOM between wet and dry seasons indicated seasonal changes in the sources of riverine DOM in relation with changes in water level and connectivity with wetlands/floodplains. The high $SUVA_{254}$ and low $S_R$ values during the wet seasons indicate the mobilisation of fresh aromatic DOM of high molecular weight due to the increased water flow through DOM-rich upper soil horizons during high flow periods (Striegl et al., 2005; Neff et al., 2006; Mann et al., 2012; Bouillon et al., 2014). Wetlands and floodplains were the main sources of terrestrial DOM at the basin scale during wet seasons, as shown by the relationships between DOC and wetland extent (Fig. 7). Among the different terrestrial humic-like components, C4 was the most affected by fluctuations in the connectivity with wetlands/floodplains. The increase in the relative contribution of C4 suggests that this component was mobilized in greater proportion relative to others (Fig. 5). This observation is consistent with a recent study conducted in boreal streams, in which a component similar to C4 was found to increase





relative to other humic-like fluorophores (equivalent to C1 and C2) in stream waters during
the peak spring melt due to the higher abundance of this component in uppermost soil
horizons of wetlands (Kothawala et al., 2015). The longitudinal and seasonal variations in
%C4 in the upper Zambezi are consistent with the hypothesis that C4 is mainly produced
in the upper soil horizons of wetlands/floodplains and therefore preferentially mobilized
during high flow periods.
**4.2.2 WRT modulates the downstream patterns of DOM.** During the dry season, DOM
was characterized by lower $SUVA_{254}$ and higher $S_R$ values, indicating the transport of
compounds of lower aromaticity and lower average molecular weight compared to high
flow periods. The difference in downstream gradients of DOM compared to the wet
seasons can be explained in part by the loss of connectivity between rivers and riparian
wetlands/floodplains and the deepening of hydrological flowpaths through DOM-poor
deeper subsoil horizons during the dry season (e.g. Striegl et al., 2005; Bouillon et al.,
2014). Changes of connectivity with wetland during the dry season was also found to
strongly impact $CO_2$ and $CH_4$ distribution in the Zambezi (Teodoru et al., 2015). That being
said, the considerable decrease in water discharge during dry/base flow period compared
to wet/high flow periods (Fig. 2) likely leads to a decrease in water velocities and
subsequently to an increase in solutes residence time, allowing a more efficient
degradation of terrestrial DOM along a given section. For illustration, the preferential
downstream loss of $a_{350}$ compared to DOC in the upper Zambezi, associated with a
gradual decrease of $SUVA_{254}$ and increase of $S_R$, is a strong evidence of the preferential
loss of the terrestrial and aromatic fraction of DOM through photodegradation (e.g.
Spencer et al., 2009; Weyhenmeyer et al., 2012). The stable level of $F_{Max}$ of C3 suggests
a continuous supply of this component, likely due to microbial degradation of terrestrial



DOM. In addition, the increase in WRT could favour the accumulation of DOM from
autochthonous sources as suggested by higher values of FI and the gradual increase in
$F_{Max}$ for C5 (Fig. 3 and 4). Flushing during high flow periods perturbs the downstream
gradient of DOM established during base flow because (1) increase in water level
mobilizes a greater proportion of terrestrial DOM and (2) increase in water velocities
increases the travel distance of humic and aromatic terrestrial compounds before removal
due to microbial and photochemical degradation processes and limits the accumulation of
autochthonous DOM in the water column.
**4.2.3. Retention of water by lakes/reservoirs.** Longitudinal patterns of DOM were
affected by the presence of reservoirs independently of water level fluctuations, in which
DOM was characterized by low aromaticity and molecular weight and higher microbial
contribution (Fig. 4 and 6). The net loss of DOC and the preferential loss of the coloured
and aromatic fraction of DOM (based on $a_{350}$ and $SUVA_{254}$, Fig. 3) in lakes and reservoirs
have been previously documented (Hanson et al., 2011; Köhler et al., 2013) and attributed
to the combination of several processes including flocculation, photochemical and
microbial degradation (Cory et al., 2007; von Wachenfeldt and Tranvik, 2008; Köhler et
al., 2013; Kothawala et al., 2014). Although we were not able to estimate the relative
contribution of these mechanisms, our results indicate that the humic-like fractions of DOM
(C1-C4) were more susceptible to degradation compared to the protein-like fraction (C5),
an observation consistent with recent studies carried out in boreal lakes (Kothawala et al.,
2014). The level of fluorescence of C5 could be additionally sustained by the FDOM from
primary producers such as macrophytes (Lapierre and Frenette, 2009), that also lead to
low values of the partial pressure of $CO_2$ in the Kariba and Cahora Bassa reservoirs
(Teodoru et al., 2015).



In agreement with others studies (e.g. Hanson et al., 2011), the effects of reservoirs
on the fate of DOM were related to their specific WRT. The Itezhi Tezhi Reservoir had
little effect on longitudinal patterns of DOM, as also suggested by a recent study (Zürbrugg
et al., 2013), likely due to its relatively low WRT (0.7 yr, Kunz et al., 2011) compared to
the Kariba (5.7 yr, Magadza, 2010) and the Cahora Bassa (~2 yr, Davies et al., 2000)
reservoirs. The DOC concentrations upstream and downstream of the Cahora Bassa
Reservoir were similar but DOM composition exhibited significant changes within the
reservoir compared to upstream and downstream stations, suggesting a balance between
loss and production of new compounds. In fact, the Kariba Reservoir was the most
important reservoir responsible for the perturbation of the longitudinal DOM gradient. The
seasonal variability of DOM at the outlet of the Kariba Reservoir, both in terms of
concentration and composition, was drastically reduced compared to the seasonal
patterns observed in the upper Zambezi (Fig. 3 and 5). This was also illustrated by data
from a two-years monitoring of the Zambezi River 70 km downstream of the Kariba Dam,
showing that the terrestrial fraction of DOM leaving the reservoir has undergone extensive
transformation (Table 2).
Beyond their role as hotspots for DOM processing and mineralization,
lakes/reservoirs act as a hydrological buffer and reduce the temporal variability of
downstream water flow (Goodman et al., 2011; Lottig et al. 2013). Except for some
isolated events, water discharge remained constant at Kariba Dam due to hydropower
management (Fig. 2). Combined with the low temporal variability in DOM content (Table
2), DOC fluxes at the outlet of the Kariba Reservoir were relatively invariant and ranged
between $8.3 \times 10^7$ and $9.7 \times 10^7$ kg yr$^{-1}$. This results in a twofold decrease of DOC fluxes



during the wet seasons between upstream inputs from the upper Zambezi and export at the outlet of the Kariba Reservoir, but in the increase by a factor of 12 during the dry season (Fig. 8). On a longitudinal perspective, lakes/reservoirs can thus shift from DOM source to sink relative to upstream ecosystems while reducing the temporal variation of DOM fluxes and composition to downstream ecosystems. That being said, DOM losses were largely offset during the wet seasons by inputs from the Kafue and the Shire rivers as well as from wetlands in the lower Zambezi (Fig. 3 and 8). Therefore, the spatial arrangement of the different elements that constitute large river networks such as lakes/reservoirs, wetlands/floodplains and large tributaries is a key aspect in controlling DOM export at the basin scale.

**4.3. Comparison with others rivers.** The results of this study are similar to those reported in large rivers from other biomes regarding (1) the role of peak flow periods in exporting a greater portion of terrestrial aromatic and humic DOM (Neff et al., 2006; Duan et al., 2007; Holmes et al., 2008; Walker et al., 2013), (2) the disproportionate importance of riparian wetlands and floodplains in regulating in-stream chemistry (Battin, 1998; Hanley et al., 2013; Borges et al., 2015b) and (3) the reactivity of terrestrial DOM during its transport (Massicotte and Frenette, 2011; Cawley et al., 2012; Wehenmeyer et al., 2012). However, while changes in temperature have been suggested as a secondary factor impacting DOM patterns in temperate and boreal streams and rivers (Kothawala et al., 2014; Raymond et al., 2015), changes in longitudinal DOM patterns in the Zambezi Basin were only controlled by changes in hydrology. Indeed, water temperatures were systematically elevated with values mainly ranging from 25 to 29°C (data not shown) and no significant patterns were apparent between the contrasting seasons.



Our study clearly illustrates that the DOC in a given station is the legacy of

upstream sources and their degree of processing during transport, and suggests that WRT
is a major driver controlling the fate of DOM in freshwaters (the latter resulting from the
competition between transport and degradation processes). Seasonal changes in DOM
concentration and composition in large rivers assessed by monitoring programs are often
explained by vertical changes in DOM sources mobilized during high flow and base flow
conditions, i.e. shallow versus deep sources along the soil profile (Neff et al., 2006; Mann
et al., 2012; Bouillon et al., 2014). Our results show that the upstream degradation history
of DOM during transit should also be taken into consideration, especially during base flow
periods. Given the strong reactivity of fresh terrestrial humic DOM exported during high
flow periods (e.g. Holmes et al., 2008; Mann et al., 2012) and the ability of large
hydrological events to transport DOM downstream over large distances (Raymond et al.,
2015), the functioning of large rivers at the seasonal scale and their impacts on receiving
ecosystems (e.g. coastal waters) should deserve more attention.

**Author contributions**

The research project was designed by AVB and SB, field data collection was done by
CRT and FCN. CDOM and FDOM measurements were done by TL with the help of FD.
Data analysis was done by TL with the help of PM for PARAFAC modelling. Manuscript
was drafted by TL that was commented, amended and approved by all co-authors.

**Acknowledgements**

This work was funded by the European Research Council (ERC-StG 240002 AFRIVAL),
the Fonds National de la Recherche Scientifique (FNRS, FluoDOM J.0009.15), the



Research Foundation Flanders (FWO-Vlaanderen), the Research Council of the KU
Leuven. We thank Christiane Lancelot (Université Libre de Bruxelles) for access to the
Perkin-Elmer UV/Vis 650S. TL is a postdoctoral researcher at the FNRS. AVB is a senior
research associate at the FNRS.
**Supplementary Information** accompanies this paper.

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

**Figure captions**
**Figure 1 –** Map of the Zambezi basin illustrating the digital elevation model, wetlands and
floodplains areas (data from Lehner and Döll, 2004), the main hydrological network and
the distribution of sampling sites along the Zambezi and the Kafue rivers.
**Figure 2 –** Water discharge between January 2012 and January 2014 for (a) the Zambezi
River at Victoria Falls and at Kariba Dam, and (b) for the Kafue River at Hook Bridge
located upstream of the Itezhi Tezhi Reservoir and at the Kafue Gorge Dam (data from
Zambia Electricity Supply Corporation Limited, ZESCO). Bars refer to the three periods
during which field campaigns were performed.
**Figure 3 –** Longitudinal variations of DOM properties along the Zambezi River (left panels)
and the Kafue River (right panels) during the wet and the dry seasons. From top to bottom
the panels represent: DOC, $a_{350}$, $\delta^{13}C_{DOC}$, SUVA$_{254}$, S$_R$ and FI. Dark gray and light gray
rectangles in background represent the approximate position along the mainstream of



wetlands/flooplains areas and reservoirs, respectively. Roman numerals refer to (I) Barotse Floodplain, (II) Chobe Swamps, (III) Kariba Reservoir, (IV) Cahora Bassa Reservoir, (V) lower Zambezi wetlands for the Zambezi River and (VI) Lukanga Swamps, (VII) Itezhi Tezhi Reservoir and (VIII) Kafue Flats for the Kafue River. The diamonds represent samples collected from main tributaries upstream to their confluence with mainstreams: (IX) the Kabompo, (X) the Kafue, (XI) the Luangwa, (XII) the Mazoe and (XIII) Shire River for the Zambezi River and (XIV) the Lunga River for the Kafue River. Symbols and error bars for data collected during the wet seasons represent the average and standard deviation between the two field campaigns performed in 2012 and 2013, respectively.

**Figure 4 –** Longitudinal variations of FDOM along the Zambezi River (left panels) and the Kafue River (right panels) during the wet and the dry seasons. From top to bottom the panels represent: $F_{Tot}$ and $F_{Max}$ for each PARAFAC component. The diamonds represent samples taken from main tributaries upstream their confluence with mainstreams.

**Figure 5 –** Longitudinal variations of the relative contribution of PARAFAC component along the Zambezi River (left panels) and the Kafue River (right panels) during the wet and the dry seasons. The diamonds represent samples taken from main tributaries upstream their confluence with mainstreams.

**Figure 6 –** Graphical representation of PCA results, including loadings plot for the input variables and scores plot for water samples collected during the wet dry (circles) and the wet (triangles) seasons. Water samples from the Zambezi River (ZBZ) were classified





according to its source and the three major segments of the Zambezi basin. Samples from
reservoirs (i.e. Kariba, Cahora Bassa and Itezhi Tezhi reservoirs) were classified together.

**Figure 7 –** Relationships between DOC and % Wetlands in the Zambezi and the Kafue
rivers, with *:$p<0.1$, and ***:$p<0.001$.

**Figure 8 –** DOC fluxes calculated at different locations along the Zambezi River during
the wet and the dry seasons. Vertical arrows represent changes in DOC fluxes at a same
location between the wet and the dry seasons. Diagonal changes represent longitudinal
variations.





**Table 1–** Spectral characteristics of the five fluorophores identified by PARAFAC modelling, correspondence with previously
identified components in different environments, general assignment and possible source. Numbers in brackets refer to the
second peak of maximal excitation.

| Component | Maximum Excitation (nm) | Maximum Emission (nm) | Comparison with others environments | | | | | | | | Assignement | Possible source[a] |
|---|---|---|---|---|---|---|---|---|---|---|---|---|
| | | | St Lawrence River[1] | Large Arctic rivers[2] | Boreal Lakes[3,4] | Subtropical wetlands[5,6] | Tropical wetland[7] | Temperate estuary[8] | Coastal waters[9] | Marine waters[10] | | |
| C1 | <240 (325) | 443 | C2 | C1 | C4 | C1 | C1 | C4 | — | C1 | Terrestrial humic-like | T |
| C2 | <240 (365) | 517 | C3 | C3 | C3 | C5 | C4 | C2 | C3 | — | Terrestrial humic-like | T |
| C3 | <240 (305) | 383 | C7 | — | C2 | C4 | C3 | C6 | C6 | C4 | Microbial humic-like | Au[9],M[3,7,10], An[8] |
| C4 | <240 | 405 | C1 | — | C5 | C2 | C2 | C1 | C1 | — | Terrestrial humic-like | T[5-6,8], P[1,4] |
| C5 | 275 (<240) | 337 | C4 | C5 | C6 | C8 | — | C7 | C4 | C2 | Tryptophan-like | Au[1,9], M[2,8] |

[a] T: Terrestrial inputs; Au: Autochthonous primary production; An: Anthropogenic origin; M: Microbial degradation; P: Photochemical degradation.

1) Massicotte and Frenette (2011); 2) Walker et al. (2013); 3) Kothawala et al. (2014); 4) Kellerman et al. (2015); 5) Yamashita et al. (2010); 6) Cawley et al. (2012); 7) Zürbrugg et al. (2013); 8) Stedmon and Markager (2005); 9) Yamashita et al. (2008); 10) Jørgensen et al. (2011).



**Table 2 –** Temporal variations of DOM properties measured at the outlet of the Kariba Reservoir during a one year and half
monthly sampling (from February 2012 to November 2013).

| | DOC (mg L-1) | $\delta^{13}C_{DOC}$ (‰) | $a_{350}$ (m-1) | $SUVA_{254}$ (L mgC-1 m-1) | $S_R$ | %C1 | %C2 | %C3 | %C4 | %C5 |
|---|---|---|---|---|---|---|---|---|---|---|
| Min | 2,00 | -23,96 | 1,00 | 1,39 | 1,010 | 27,7 | 12,2 | 16,1 | 4,0 | 12,3 |
| Max | 2,60 | -22,26 | 2,50 | 3,11 | 1,428 | 36,5 | 16,6 | 26,2 | 13,8 | 35,9 |
| Mean | 2,22 | -23,08 | 1,60 | 2,02 | 1,185 | 34,1 | 15,2 | 24,1 | 9,3 | 17,3 |
| S.D. | 0,17 | 0,37 | 0,44 | 0,43 | 0,141 | 2,4 | 1,2 | 2,7 | 3,1 | 6,2 |
| *n* | 20 | 20 | 12 | 12 | 12 | 12 | 12 | 12 | 12 | 12 |






**Figure 1**

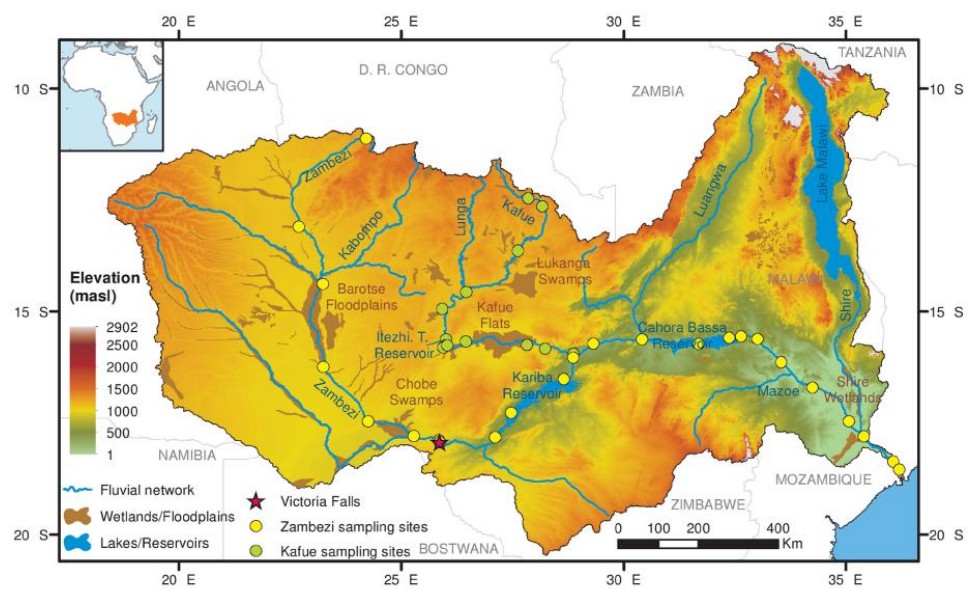






**Figure 2**

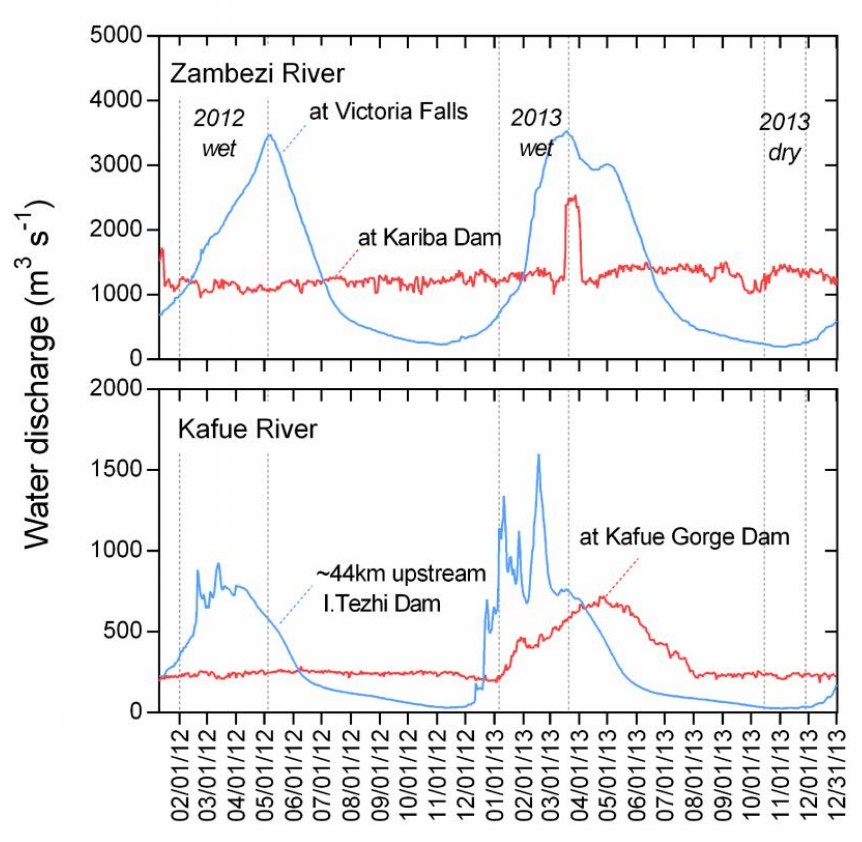


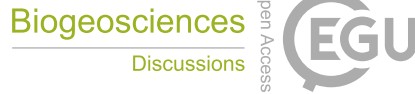



**Figure 3**

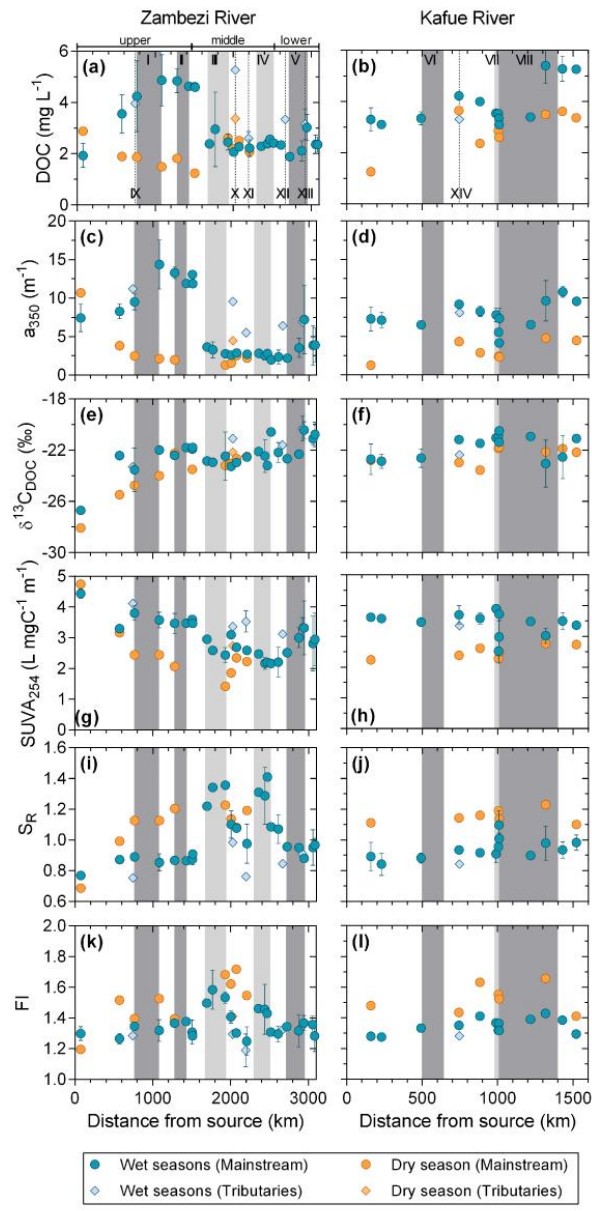






**Figure 4**

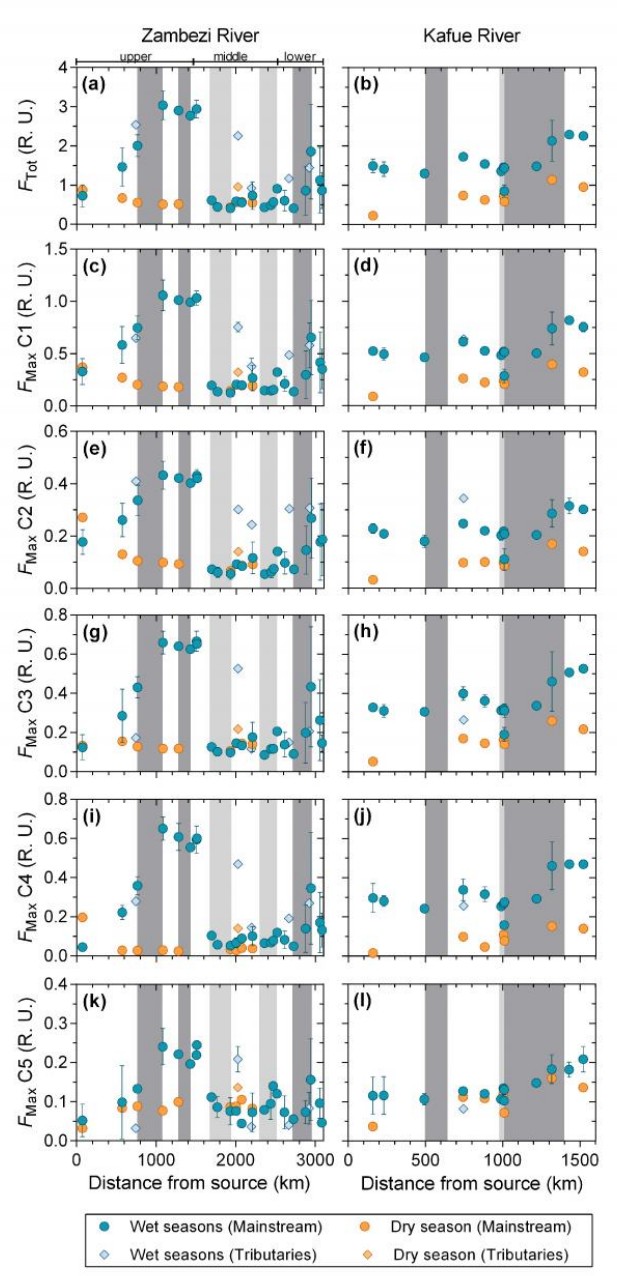




**Figure 5**

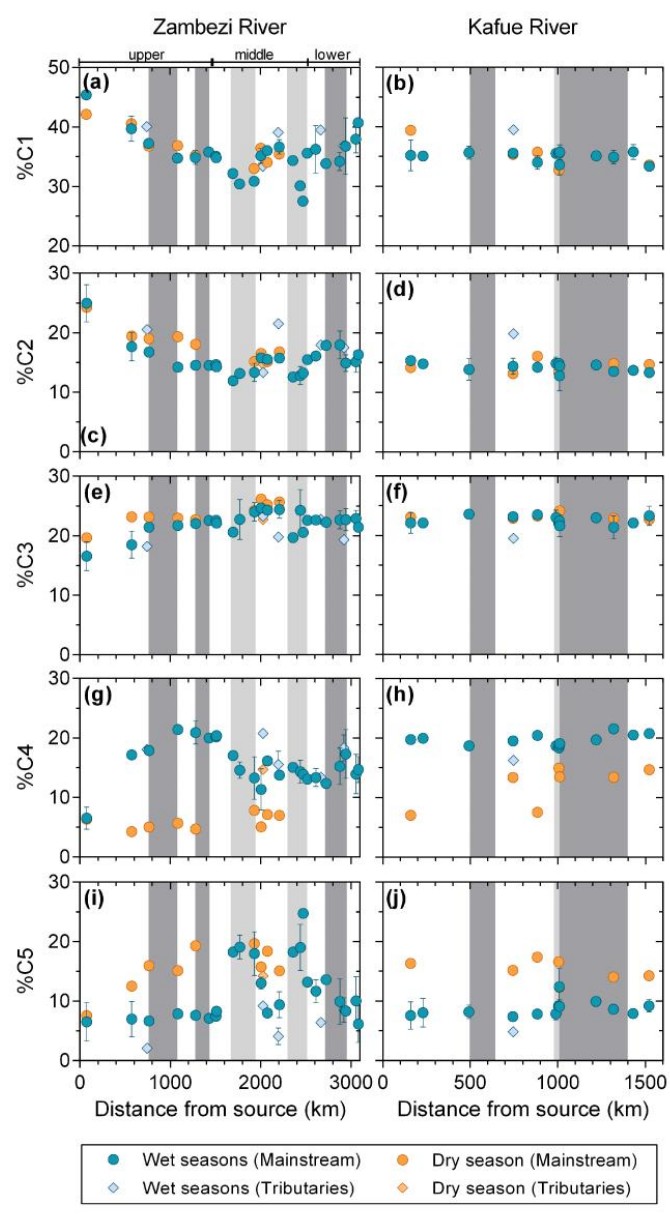






**Figure 6**

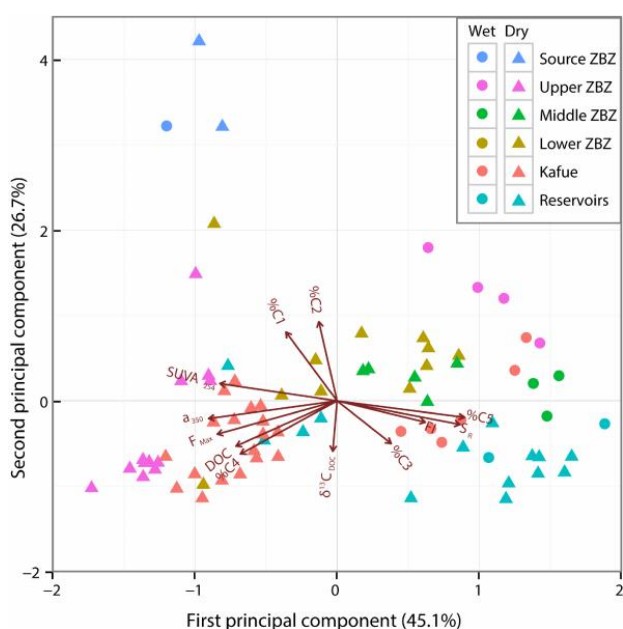




**Figure 7**

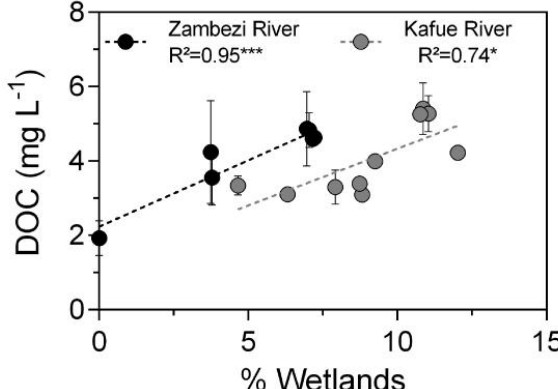






**Figure8**

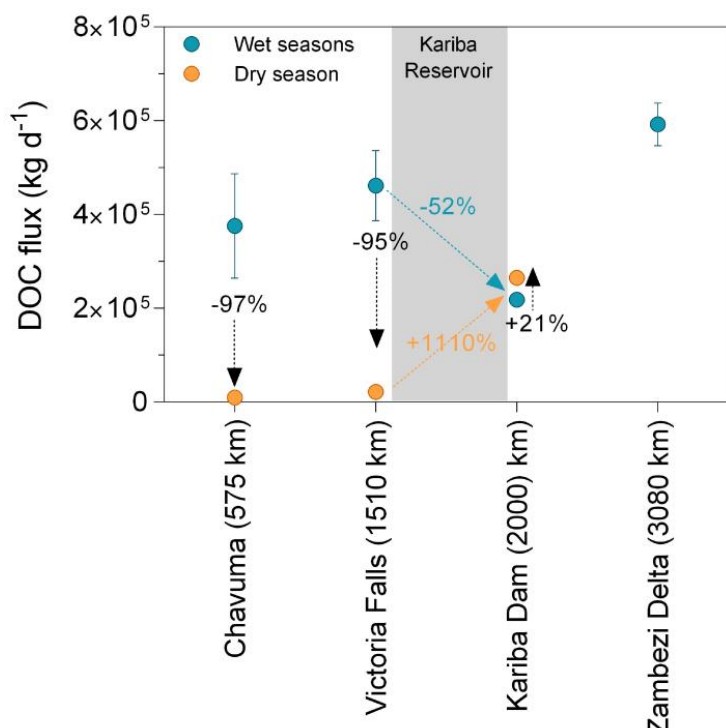



