# Peer review of "Along-stream transport and transformation of dissolved organic"

_Biogeosciences, 2016_

## Referee Comment (RC1) · Anonymous Referee #1 · 7 Mar 2016

The submitted MS presents a detailed analysis of transport and transformation of DOM along the main stem of the Zambesi and its largest tributary. A particular focus is put on the effects of floodplains/wetlands and reservoirs as well as low-flow vs high-flow conditions on the longitudinal patterns in DOM concentration and composition. It is the first study to present such a detailed analysis for a whole, large river system, and in particular for a tropical river other than the Amazon. Thus, the subject of the study will be of interest for the readership of Biogeosciences. Methods and results are presented in a clear, comprehensive way. The discussion features a satisfying review of the literature and compares results of this study to the state of the art in that field. The manuscript is well written and tables and figures are mainly of good quality. I suggest publication of the MS after minor revisions.

Major comment:

**BGD**

In addition to spectral properties, the authors measured delta13C of the DOM. They present the results, but they do not interpret and discuss the values. I suggest that the authors include a short interpretation of these delta13C values based on an isotopic mixing model to estimate the proportions of different terrestrial and autochtonous sources.

General comments

L100: Maybe I am wrong, but wouldn't that rather be a unimodal distribution? Bimodal would mean that there is a second maximum. Is there a second, smaller maximum? If yes, please clarify.

L114-118 & L121-123: Please, give a reference for these values (volumes and surface areas).

L150-151: Please, replace 'most cases' by a number of cases or the percentage. Or report e.g. the 95th percentile of the reproducibility.

Section 2.6: You should start this section with one to two sentences explaining what the aim of this PCA is.

Section 3.1: When you describe the longitudinal and seasonal patterns of all these indices, you should shortly repeat what each of these indices indicates. That would increase the comprehensibility for the broader readership. That is in particular true for the delta13C values. Here, you should maybe cite some typical end-member values.

L350-351: Where do you show the correlation between dominant land cover and DOM gradients?

L355: You should discuss the delta13C values. What does a low delta13C indicate? What are the endmembers?

Figure 3: Overall, the figures are of a very good quality. However, in Figure 3, at least when printed, it is hard to distinguish between the numbers I, II, III.

[Figure]

[Figure]

---

## Referee Comment (RC2) · Anonymous Referee #2 · 11 Mar 2016

Comments to Author

Summary: In this manuscript the authors present new DOC and DOM composition data from one of the World's largest tropical rivers: the Zambezi River. Samples were collected during both dry and wet seasons and along the river and one of its tributaries. The results indicated clear seasonal differences in sources and processing of DOM as well as down-river shifts in concentrations and composition. "Humic"-like DOM dominated in headwaters close to forests and at wet conditions when wetlands were dominating sources of DOM. In contrast, at dry conditions the DOM composition shifted towards more aquatically produced, or influenced, material. The authors claim that these differences are primarily driven by shifts in discharge, which influences connectivity with e.g. wetlands, and water residence times. As has been noted before, the effect of reservoirs or lakes have a particularly significant role in increasing water

residence times and thereby DOM composition and concentrations.

Contributions: Although the patterns presented and conclusions drawn are not revolutionary, they are indeed important since this type of data from tropical rivers is rare. In addition, the results largely confirm previous interpretations of DOM dynamics in boreal and temperate areas. This is interesting since it suggests that, although the details may differ (e.g. microbial community composition), the large-scale governing processes and functioning are similar across biomes.

The manuscript was a pleasure to read. After having reviewed several poorly written manuscripts recently, it was a joy to see a well written and logically organized text. Still, I do have some minor remarks detailed in a number of general and technical comments below.

General comments:

-The description of some of the methodology requires additions and clarifications.

-The use of some terminology is confusing (not uncommon when it comes to this type of terminology) and I suggest clarification. One clear example is the apparent dichotomy between terrestrial and microbial, which is clearly misleading since substantial portions of DOM may be of terrestrial microbial origin.

-The relationships between DOM properties and landscape characteristics is interesting, but presented in the Discussion section. I suggest the authors add a paragraph or two about these results in the Results section.

Altogether, this manuscript is a valuable addition to the scientific field and I support its publication in Biogeosciences. The science is as far as I can tell sound and well communicated. I recommend minor revisions of the manuscript before the editor considers publication of the manuscript.

Technical comments:

Abstract (why no line numbers in the abstract?)

Line 13-14: You write "terrestrial DOM dynamics shifted from transport-dominated during the wet seasons towards degradation". I don't think this terminology matches; what do you mean "towards degradation"? Do you mean that it shifted to a state dominated by in-stream processing?

Introduction

Line 41: This is only partly true. Sure, DOM composition controls reactivity but there are other factors that may be equally important. You identify one: water residence times. However, there are others as well, see e.g. Marín-Spiotta, E., K. E. Gruley, J. Crawford, E. E. Atkinson, J. R. Miesel, S. Greene, C. Cardona-Correa, and R. G. M. Spencer (2014), Paradigm shifts in soil organic matter research affect interpretations of aquatic carbon cycling: transcending disciplinary and ecosystem boundaries, Biogeochemistry, 117(2-3), 279-297, doi: 10.1007/s10533-013-9949-7.

Line 71: ultraviolet

Line 74-75: I know this terminology is common, but it is rather misleading, which I often point out. Terrestrial vs. microbial is not a dichotomy. On the contrary, much DOM from the terrestrial environment is of microbial origin. I think it is better to call them terrestrial and aquatic inputs.

Line 86-88: I don't know if the use of prepositions is correct here. I suggest changing to "...drivers of downstream patterns in DOM at the scale of a large tropical river, with a specific attention to the..." Materials and methods

Line 91: northwestern Zambia

Line 100: If it is a single peak it is not bimodal. A bimodal distribution has two peaks.

Line 103: I suggest changing the comma to a semi-colon: "...whole catchment; forests (20%)..."

Line 128-130: I suggest you move the year before the parentheses. Now you interrupt "the flow". So e.g. "...wet season 2013 (6 January to 21 March, n = 41) and dry season 2012 (..."

Line 140: what do you mean by "conditioned"?

Line 141: Did you use any blanks? I am always suspicious when filters made by organic compounds are used for DOM analyses.

Line 148: Were the DOM samples kept cold during sampling and transport? Due to logistical reasons I guess not (and you added phosphoric acid) but could be worth noting. Any potential effects of this sample handling? In addition, where were the analyses (concentrations, isotopes, FDOM, CDOM) performed? In Belgium?

Line 151: Do these uncertainty bounds include both accuracy and precision? Relative which standard are carbon isotope values reported?

Line 171-173: Again a somewhat confusing terminology. Is there a dichotomy between aromatic and hydrophobic? Is it aromatic vs. aliphatic?

Line 172: "...indicative of the presence..."

Line 193: Should this be "Raman units"?

Line 196: "The PARAFAC model was using..."

Line 197: This is repetitious so I suggest adding "Furthermore, the PARAFAC..."

Line 200: "...a two-year monitoring..."

Line 210-211: Here is terrestrial vs. microbial again. I suggest changing this terminology.

Line 214: Define PCA

Results

Line 223-224: "...one dry season; the two wet seasons' data..."?

Line 226: "...during the dry period..."

Line 262-263: Remove "a" before "maximum" and "minimum"

Line 266: "globally" seems strange here

Line 267: I guess this should read "except"

Line 277-278: Here is terrestrial vs. microbial again. "aquatic microbial" would be fine

Line 283: I found "corresponding river sections" unclear. Could you clarify?

Line 288: "as" seems out of place here. Perhaps "...downstream concurrent with DOC concentrations..."

Line 318: Do you mean "all samples during the dry season"? I found this unclear.

Line 319: what other variables?

Discussion

Line 328: Do you mean "conversely" instead of "inversely"?

Line 340-343: Perhaps, but from the figure it looks like C1, C2 and C3 are more related to PC2.

Line 348: "in" instead of "of"?

Line 356-357: "...in the northern part of the basin at the headwaters of the Zambezi to grasslands..."

Line 370-371: Aren't these results and should therefore be presented in the Results section?

Line 393: This only applies to water residence times, not necessarily solute residence times since they are dependent on vertical fluxes and in-stream recycling as well.

Line 397: Why only photodegradation? This should also include microbial degradation.

Line 403-404: "...(1) increasing water levels mobilizes a greater proportion of terrestrial DOM and (2) higher water velocities..."

Line 409: What does "in which" refer to? I found this sentence unclear.

Line 420-422: Is it more likely that this is due to macrophytes than to algae? What about CO2 evasion?

Line 434-436: This agrees with work in temperate/boreal systems, see e.g. Winterdahl, M., M. Erlandsson, M. N. Futter, G. A. Weyhenmeyer, and K. Bishop (2014), Intra-annual variability of organic carbon concentrations in running waters: Drivers along a climatic gradient, Global Biogeochemical Cycles, 28(4), 451-464, doi: 10.1002/2013GB004770.

Line 437: According to Table 2 this is a 1.5 year long monitoring.

Line 446-448: This is interesting! Could you then estimate the loss/production of C in the reservoir by using CO2 and CH4 data?

Line 450: "...sources to sinks..."

Line 461-462: See also Fiebig et al. (1990), Dosskey & Bertsch (1994) or Hinton et al. (1998).

Fiebig, D. M., M. A. Lock, and C. Neal (1990), Soil water in the riparian zone as a source of carbon for a headwater stream, Journal of Hydrology, 116(1-4), 217-237

Dosskey, M. G., and P. M. Bertsch (1994), Forest sources and pathways of organic matter transport to a blackwater stream: a hydrologic approach, Biogeochemistry, 24(1), 1-19

Hinton, M. J., S. L. Schiff, and M. C. English (1998), Sources and flowpaths of dissolved organic carbon during storms in two forested watersheds of the Precambrian Shield,

Biogeochemistry, 41(2), 175-197

Line 465-466: There are several references for this; the Winterdahl et al. (2014) paper referred to above is another.

Figure captions

Line 728: "...upstream of their..."

Line 746: Remove "wet"

Line 754: This is really exports. Fluxes are technically export per unit area.

Line 755-756: "...exports at the same location between wet and dry seasons."

Table 1: Very interesting!

Table 2 Line 763: "...during a one and a half year monthly..."

Figure 7: Are these all sites? The number of sites in the Zambezi River seems few compared to other figures. Is this a selection of sites? If so, based on what?

Figure 8: This is rather DOC export. Flux is export per unit area.

---

## Author Comment (AC1) · 23 Mar 2016

Author response to Anonymous Referee #1

The submitted MS presents a detailed analysis of transport and transformation of DOM along the main stem of the Zambesi and its largest tributary. A particular focus is put on the effects of floodplains/wetlands and reservoirs as well as low-flow vs high flow conditions on the longitudinal patterns in DOM concentration and composition. It is the first study to present such a detailed analysis for a whole, large river system, and in particular for a tropical river other than the Amazon. Thus, the subject of the study will be of interest for the readership of Biogeosciences. Methods and results are presented in a clear, comprehensive way. The discussion features a satisfying review of the literature and compares results of this study to the state of the art in that field.

The manuscript is well written and tables and figures are mainly of good quality. I suggest publication of the MS after minor revisions.

- Reply: We thank the reviewer for the positive evaluation of our manuscript and for his/her comments and suggestions.

Major comment:

In addition to spectral properties, the authors measured delta13C of the DOM. They present the results, but they do not interpret and discuss the values. I suggest that the authors include a short interpretation of these delta13C values based on an isotopic mixing model to estimate the proportions of different terrestrial and autochtonous sources.

- Reply: We have added in the revised manuscript a new section that focuses more in depth on the results of $\delta$13C of DOC (section 4.2 in the revised manuscript). First, we compared our data with previously published data from other African tropical rivers. Secondly, we discussed the possible reasons leading to the increase of values along the Zambezi mainstem. Based on the lack of marked 13C-depletion DOC in the reservoirs, we suggest that phytoplankton production has little effect on the $\delta$13C of DOC and that the increased in $\delta$13CDOC is to a large extent due to increased contribution from C4 vegetation. Finally, we performed a mass balance calculation to estimate the relative contribution of C3 and C4 plants on the DOM pool in the Zambezi basin. End-members values were fixed at -27.1 ‰ for C3 plants and -12.1 ‰ for C4 plants. The value of -27.1 ‰ was calculated in a geographical information system (ArcGIS), based on the equation of Kohn (2010) that estimates the $\delta$13C signature of C3 vegetation based on mean annual precipitation, altitude and latitude. Available and public datasets for annual rainfall (Hijmans et al., 2005) and digital elevation model (HydroSHEDs) were used. The value of -12.1 ‰ was chosen based on a study conducted in the Tana River basin (Kenya) which presents similar shift in vegetation cover (Tamooh et al., 2012). We have also added another supplementary figure that shows the spatial

variability of the estimated δ13C signature of C3 plants in the Zambezi basin. Also, the first paragraph of the section 4.3.1 (previously 4.2.1) has been slightly modified in order to avoid repetition with the previous section.

General comments

L100: Maybe I am wrong, but wouldn't that rather be a unimodal distribution? Bimodal would mean that there is a second maximum. Is there a second, smaller maximum? If yes, please clarify.

- Reply: Indeed it is a unimodal distribution. The text has been corrected.

L114-118 & L121-123: Please, give a reference for these values (volumes and surface areas).

- Reply: We have added references for each reservoir.

L150-151: Please, replace 'most cases' by a number of cases or the percentage. Or report e.g. the 95th percentile of the reproducibility.

- Reply: The percentage of samples with a reproducibility higher than 5% for DOC and 2% for δ13CDOC was lower than 5%. This precision has been added in the revised manuscript.

Section 2.6: You should start this section with one to two sentences explaining what the aim of this PCA is.

- Reply: This has been made.

Section 3.1: When you describe the longitudinal and seasonal patterns of all these indices, you should shortly repeat what each of these indices indicates. That would increase the comprehensibility for the broader readership. That is in particular true for the delta13C values. Here, you should maybe cite some typical end-member values.

- Reply: This has been made.

L350-351: Where do you show the correlation between dominant land cover and DOM gradients?

- Reply: In fact the effect of land cover and DOM gradient is discussed just below, in the section 4.3.1. In order to make the manuscript clearer, this sentence has been removed and we reworked the paragraph 4.3.

L355: You should discuss the delta13C values. What does a low delta13C indicate? What are the endmembers?

- Reply: This comment has been been addressed by adding the new section 4.2, see also our reply to earlier comments above.

Figure 3: Overall, the figures are of a very good quality. However, in Figure 3, at least when printed, it is hard to distinguish between the numbers I, II, III.

- Reply: We appreciate this comment. The figure 3 has been modified.

---

## Author Comment (AC2) · 23 Mar 2016

Author response to Anonymous Referee #2

Comments to Author Summary: In this manuscript the authors present new DOC and DOM composition data from one of the World's largest tropical rivers: the Zambezi River. Samples were collected during both dry and wet seasons and along the river and one of its tributaries. The results indicated clear seasonal differences in sources and processing of DOM as well as down-river shifts in concentrations and composition. "Humic"-like DOM dominated in headwaters close to forests and at wet conditions when wetlands were dominating sources of DOM. In contrast, at dry conditions the DOM composition shifted towards more aquatically produced, or influenced, material. The authors claim that these differences are primarily driven by shifts in discharge, which

influences connectivity with e.g. wetlands, and water residence times. As has been noted before, the effect of reservoirs or lakes have a particularly significant role in increasing water residence times and thereby DOM composition and concentrations.

Contributions: Although the patterns presented and conclusions drawn are not revolutionary, they are indeed important since this type of data from tropical rivers is rare. In addition, the results largely confirm previous interpretations of DOM dynamics in boreal and temperate areas. This is interesting since it suggests that, although the details may differ (e.g. microbial community composition), the large-scale governing processes and functioning are similar across biomes. The manuscript was a pleasure to read. After having reviewed several poorly written manuscripts recently, it was a joy to see a well written and logically organized text. Still, I do have some minor remarks detailed in a number of general and technical comments below.

General comments: -The description of some of the methodology requires additions and clarifications.

-The use of some terminology is confusing (not uncommon when it comes to this type of terminology) and I suggest clarification. One clear example is the apparent dichotomy between terrestrial and microbial, which is clearly misleading since substantial portions of DOM may be of terrestrial microbial origin.

-The relationships between DOM properties and landscape characteristics is interesting, but presented in the Discussion section. I suggest the authors add a paragraph or two about these results in the Results section.

Altogether, this manuscript is a valuable addition to the scientific field and I support its publication in Biogeosciences. The science is as far as I can tell sound and well communicated. I recommend minor revisions of the manuscript before the editor considers publication of the manuscript.

- Reply: We thank the reviewer for the positive evaluation of our manuscript and for

his/her comments and suggestions. We are also grateful to the reviewer for his/her numerous corrections and suggestions for improving the readability of the manuscript.

Technical comments: Abstract (why no line numbers in the abstract?)

- Reply: We made an error during the upload process.

Line 13-14: You write "terrestrial DOM dynamics shifted from transport-dominated during the wet seasons towards degradation". I don't think this terminology matches; what do you mean "towards degradation"? Do you mean that it shifted to a state dominated by in-stream processing?

- Reply: What we meant is that during high flow periods, the downstream transport of DOM dominates relative to degradation because of higher water velocities (i.e. lower water residence time). The situation is inversed during low flow periods because decreasing water velocities enhances the degradation of DOM during its transport. The sentence has been modified in order to clarify this point: "Thus, high water discharge promotes the transport of terrestrial DOM downstream instead of its degradation while low water discharge allows the degradation of DOM during its transport.".

Introduction Line 41: This is only partly true. Sure, DOM composition controls reactivity but there are other factors that may be equally important. You identify one: water residence times. However, there are others as well, see e.g. Marín-Spiotta, E., K. E. Gruley, J. Crawford, E. E. Atkinson, J. R. Miesel, S. Greene, C. Cardona-Correa, and R. G. M. Spencer (2014), Paradigm shifts in soil organic matter research affect interpretations of aquatic carbon cycling: transcending disciplinary and ecosystem boundaries, Biogeochemistry, 117(2-3), 279-297, doi: 10.1007/s10533-013-9949-7.

- Reply: We agree with this comment. The text has been modified to make this clarification, and the reference of Marín-Spiotta et al., (2014) has been added.

Line 71: ultraviolet

- Reply: This has been corrected.

Line 74-75: I know this terminology is common, but it is rather misleading, which I often point out. Terrestrial vs. microbial is not a dichotomy. On the contrary, much DOM from the terrestrial environment is of microbial origin. I think it is better to call them terrestrial and aquatic inputs.

- Reply: We have modified the sentence as follow: "terrestrial versus aquatic microbial inputs".

Line 86-88: I don't know if the use of prepositions is correct here. I suggest changing to "...drivers of downstream patterns in DOM at the scale of a large tropical river, with a specific attention to the..."

- Reply: This has been corrected.

Materials and methods

Line 91: northwestern Zambia

- Reply: This has been corrected.

Line 100: If it is a single peak it is not bimodal. A bimodal distribution has two peaks.

- Reply: Indeed the hydrological regime of the Zambezi is unimodal. This has been corrected.

Line 103: I suggest changing the comma to a semi-colon: ...whole catchment; forests (20%)..."

- Reply: This has been corrected.

Line 128-130: I suggest you move the year before the parentheses. Now you interrupt "the flow". So e.g. "...wet season 2013 (6 January to 21 March, n = 41) and dry season 2012 (..."

- Reply: This has been corrected.

Line 140: what do you mean by "conditioned"?

- Reply: This means the preservation of the samples for the different analyses, e.g. the addition of H3PO4 for samples for DOC measurements. The text has been modified for clarity.

Line 141: Did you use any blanks? I am always suspicious when filters made by organic compounds are used for DOM analyses.

- Reply: No blanks were used on the field. However, the filters were rinsed with at least 100 ml prefiltered sample water (which was collected for analysis of total alkalinity) before collection of the DOC samples in order to "flush" the potential amount of DOC present in filters.

Line 148: Were the DOM samples kept cold during sampling and transport? Due to logistical reasons I guess not (and you added phosphoric acid) but could be worth noting. Any potential effects of this sample handling? In addition, where were the analyses (concentrations, isotopes, FDOM, CDOM) performed? In Belgium?

- Reply: The DOM samples were processed within 10 minutes of water collection and sampling bottles were kept away from direct sunlight. It was not possible to keep the samples d cold uring the transport to Belgium where the analysis were performed. However, the filtration through 0.2 $\mu$m, the addition of H3PO4, and storage in the dark should guarantee good preservation of DOM concentration and composition during the storage. This has been verified previously on samples from the Oubangui River, analyzed immediately after fieldwork and after several months of dark storage at room temperature (Bouillon et al., 2014); and also clearly illustrated when comparing CDOM properties for the Zambezi sample set: the same sample analyzed upon return in Belgium (red line) and after 3 months of storage at room temperature (blue line) give identical results (Figure 1), with differences in optical values (a350, SUVA ad SR) less than 3%.

Line 151: Do these uncertainty bounds include both accuracy and precision? Relative which standard are carbon isotope values reported?

- Reply: The uncertainty bounds correspond to precision, the word "reproducibility" was replaced by "precision". Text now reads: Quantification and calibration was performed with series of standards prepared in different concentrations, using both IAEA-C6 ($\delta$13C = -10.4 ‰ and in-house sucrose standards ($\delta$13C=-26.99 ‰. All data are reported in the $\delta$ notation relative to VPDB (Vienna Pee Dee Belemnite)"

Line 171-173: Again a somewhat confusing terminology. Is there a dichotomy between aromatic and hydrophobic? Is it aromatic vs. aliphatic?

- Reply: The text has been modified for clarity: "The SUVA254 was used as an indicator of the aromaticity of DOC with high values (>3.5 l mgC-1 m-1) indicating the presence of more complex aromatic moieties and low values (<3 l mgC-1 m-1) indicative the presence of more aliphatic compounds (Weishaar et al., 2003)."

Line 172: ". . .indicative of the presence. . ."

- Reply: This has been corrected.

Line 193: Should this be "Raman units"?

- Reply: Indeed. This has been corrected.

Line 196: "The PARAFAC model was using. . ."

- Reply: This has been corrected: "PARAFAC model was build using. . ."

Line 197: This is repetitious so I suggest adding "Furthermore, the PARAFAC. . ."

- Reply: The sentence has been reworked to avoid repetition: "Validation of the PARAFAC model was performed by split-half analysis and random initialization"

Line 200: ". . .a two-year monitoring. . ."

- Reply: This has been corrected.

Line 210-211: Here is terrestrial vs. microbial again. I suggest changing this terminology.

- Reply: We added the term "aquatic microbial DOM" in the revised version.

Line 214: Define PCA

- Reply: This has been done.

Results Line 223-224: "...one dry season; the two wet seasons' data..."?

- Reply: We have made two distinct sentences in the revised manuscript.

Line 226: "...during the dry period..."

- Reply: This has been corrected.

Line 262-263: Remove "a" before "maximum" and "minimum"

- Reply: This has been corrected.

Line 266: "globally" seems strange here

- Reply: We have replaced "globally" by "generally".

Line 267: I guess this should read "except"

- Reply: Yes. This has been corrected.

Line 277-278: Here is terrestrial vs. microbial again. "aquatic microbial" would be fine

- Reply: This has been corrected.

Line 283: I found "corresponding river sections" unclear. Could you clarify?

- Reply: The "corresponding river sections" refer to the sections of the river that crosses wetlands and floodplains. We modified the text in order make this sentence clearer: "FMax of the C4 component presented the higher percentage of increase compared to the other component in river sections flowing through wetlands/floodplains in the upper and lower Zambezi."

Line 288: "as" seems out of place here. Perhaps "...downstream concurrent with DOC

concentrations. . ."

- Reply: This has been corrected according to the suggestion of the reviewer.

Line 318: Do you mean "all samples during the dry season"? I found this unclear.

- Reply: We referred only to the samples collected in the middle and lower Zambezi. The sentence has been modified: "Samples collected in the middle and lower Zambezi during both the wet and dry seasons. . ."

Line 319: what other variables?

- Reply: The other variables are those used in the PCA, i.e. the DOM concentration (DOC concentration) and composition (isotopic and optical proxies). The sentence has been modified: ". . .defined by PARAFAC components and DOM concentration and composition.".

Discussion

Line 328: Do you mean "conversely" instead of "inversely"?

- Reply: Yes. The sentence has been modified.

Line 340-343: Perhaps, but from the figure it looks like C1, C2 and C3 are more related to PC2.

- Reply: We agree that C1 and C2 seem to be opposite to C3 along the PC2 also. However, we found that C1 and C2 are much opposed to $\delta$13CDOC along this axis. Considering the effect of the vegetation gradient on $\delta$13CDOC values (see the new section 4.2) and the fact that C1 and C2 are highest at the source of the Zambezi, this suggests that changes in land cover control the distribution of samples along PC2. We have added 2 sentences at the end of the section 4.1 to discuss this point.

Line 348: "in" instead of "of"?

- Reply: Yes. The sentence has been modified.

Line 356-357: "...in the northern part of the basin at the headwaters of the Zambezi to grasslands..."

- Reply: Following the major comment of the other reviewer regarding the lack of interpretation for $\delta$13CDOC values, a new section has been added where we discussed about the transition of land cover along the Zambezi River. This sentence has been removed in order to avoid repetition with the new section.

Line 370-371: Aren't these results and should therefore be presented in the Results section?

- Reply: We have moved this figure in the Results section as suggested. We have also added a sentence in the text to introduce this figure. The numbering of figures 4-8 have been checked in the revised manuscript.

Line 393: This only applies to water residence times, not necessarily solute residence times since they are dependent on vertical fluxes and in-stream recycling as well.

- Reply: Indeed. We have made the correction.

Line 397: Why only photodegradation? This should also include microbial degradation.

- Reply: According to the literature and personal unpublished experimental data, the preferential losses of a350 compared to DOC associated with a decrease in SUVA254 and increase in SR values are the typical expression of losses of DOM by photodegradation. Even if microbial degradation is capable of degrading aromatic compounds of terrestrial DOM, this degradation pathway is not expected to have a similar impact on DOM composition. Please note however that the microbial degradation of DOM is also taken into account in the next sentence.

Line 403-404: "... (1) increasing water levels mobilizes a greater proportion of terrestrial DOM and (2) higher water velocities..."

- Reply: This has been corrected.

Line 409: What does "in which" refer to? I found this sentence unclear.

- Reply: We replaced "in which" by "where" and moved "independently of water level fluctuations" at the end of the sentence.

Line 420-422: Is it more likely that this is due to macrophytes than to algae? What about CO2 evasion?

- Reply: We are not able to estimate precisely the role of macrophytes or algae due to the lack of adequate measurements. Therefore, we have modified the sentence in the revised manuscript to include a potential effect of algae. We have also provided more details regarding the difference between CO2 concentrations in reservoirs and rivers. The modified sentence is "The level of fluorescence of C5 could be additionally sustained by the FDOM from primary producers such as macrophytes (Lapierre and Frenette, 2009) or phytoplankton (Yamashita et al., 2008), that also lead to low values of the partial pressure of CO2 below atmospheric equilibrium in the Kariba and Cahora Bassa reservoirs while rivers (i.e., excluding reservoirs) displayed CO2 supersaturated conditions with respect to atmospheric equilibrium (Teodoru et al., 2015).".

Line 434-436: This agrees with work in temperate/boreal systems, see e.g. Winterdahl, M., M. Erlandsson, M. N. Futter, G. A. Weyhenmeyer, and K. Bishop (2014), Intra-annual variability of organic carbon concentrations in running waters: Drivers along a climatic gradient, Global Biogeochemical Cycles, 28(4), 451-464, doi:10.1002/2013GB004770.

- Reply: This reference has been added. "The role of lakes/reservoirs in lowering the seasonality of DOC in river network has also been evidenced in temperate and boreal streams and rivers in Sweden (Winterdahl et al., 2014)."

Line 437: According to Table 2 this is a 1.5 year long monitoring.

- Reply: In fact, it is a 21 month long monitoring. We have modified the text as follow: "...data from an almost two-year monitoring", and also in the Material and Methods

section.

Line 446-448: This is interesting! Could you then estimate the loss/production of C in the reservoir by using CO2 and CH4 data?

- Reply: We thank the reviewer for the valuable suggestion. Please note that in the reservoirs CO2 (and CH4) will be also affected by phytoplanktonic primary production as testified by the reported under-saturation of CO2 (Teodoru et al. 2015). This will complicate any mass balance budgets, and the available data does not allow us to investigate this, since the aim of the project was to describe the biogeochemistry in all aquatic compartments, not addressing in detail the C processing rates in specific environments, as reservoirs. Line 450: "...sources to sinks..." Reply: This has been modified.

Line 461-462: See also Fiebig et al. (1990), Dosskey & Bertsch (1994) or Hinton et al. (1998). Fiebig, D. M., M. A. Lock, and C. Neal (1990), Soil water in the riparian zone as a source of carbon for a headwater stream, Journal of Hydrology, 116(1-4), 217-237 Dosskey, M. G., and P. M. Bertsch (1994), Forest sources and pathways of organic matter transport to a blackwater stream: a hydrologic approach, Biogeochemistry, 24(1), 1-19. Hinton, M. J., S. L. Schiff, and M. C. English (1998), Sources and flowpaths of dissolved organic carbon during storms in two forested watersheds of the Precambrian Shield, Biogeochemistry, 41(2), 175-197

- Reply: All these references have been included in the revised manuscript.

Line 465-466: There are several references for this; the Winterdahl et al. (2014) paper referred to above is another.

- Reply: We have included this reference.

Figure captions

Line 728: "...upstream of their..."

- Reply: This has been modified.

Line 746: Remove "wet"

- Reply: This has been modified.

Line 754: This is really exports. Fluxes are technically export per unit area.

- Reply: This has been modified.

Line 755-756: "...exports at the same location between wet and dry seasons."

- Reply: This has been modified.

Table 1: Very interesting!

- Reply: Thank you!

Table 2 Line 763: "...during a one and a half year monthly..."

- Reply: We have modified as follow: "...during an almost two-year monthly sampling".

Figure 7: Are these all sites? The number of sites in the Zambezi River seems few compared to other figures. Is this a selection of sites? If so, based on what?

- Reply: This relationship has been obtained by considering only samples collected during the wet periods, i.e. when the hydrological connectivity between the mainstem rivers and wetlands are strong. Also, for the Zambezi, only the samples collected in the upper part of the basin have been considered due to the effect of the Kariba and Cahora Bassa reservoirs on the longitudinal pattern of DOC concentrations. These points have been added in the caption as well as in the text (Results section).

Figure 8: This is rather DOC export. Flux is export per unit area.

- Reply: The figure has been modified.

[Figure]

**FIGURE 1**

**Fig. 1.**